



# Radar and Environment-based Hail Damage Estimates using Machine Learning

Luis Ackermann[1], Joshua Soderholm[1], Alain Protat[1], Rhys Whitley[2], Lisa Ye[2], and Nina Ridder[2]

[1]Australian Bureau of Meteorology, Melbourne, Victoria 3001, Australia
[2]Suncorp Group Limited, Brisbane, Queensland 4000, Australia

**Correspondence:** Luis Ackermann (luis.ackermann@bom.gov.au)

**Abstract.** Large hail events are typically infrequent, with significant time gaps between occurrences at specific locations. However, when these events do happen, they can cause rapid and substantial economic losses within a matter of minutes. Therefore, it is crucial to have the ability to accurately observe and understand hail phenomena to improve the mitigation of this impact. While in-situ observations are accurate, they are limited in number for an individual storm. Weather radars, on the other

hand, provide a larger observation footprint, but current radar-derived hail size estimates exhibit low accuracy due to horizontal advection of hailstones as they fall, the variability of hail size distributions (HSD), complex scattering and attenuation, and mixed hydrometeor types. In this paper, we propose a new radar-derived hail product that is developed using a large dataset of hail damage insurance claims and radar observations. We use these datasets coupled with environmental information to calculate a Hail Damage Estimate (HDE) using a deep neural network approach aiming to quantify hail impact, with a critical

success index of 0.88 and a coefficient of determination against observed damage of 0.78. Furthermore, we compared HDE to a popular hail size product (MESH), allowing us to identify meteorological conditions that are associated with biases on MESH. Environments with relatively low specific humidity, high CAPE and CIN, low wind speeds aloft and southerly winds at ground are associated with a negative MESH bias, potentially due to differences in HSD or mixed hydrometeors. In contrast, environments with low CAPE, high CIN, and relatively high specific humidity aloft are associated with a positive MESH bias.

## 1  Introduction

Hail is a weather phenomenon that can cause substantial damage to crops, infrastructure, buildings, and motor vehicles (Gunturi and Tippett, 2017; Prein and Holland, 2018). It is crucial to accurately quantify and predict hail damage to enable farmers, insurance companies, and government agencies to make informed decisions and minimize the impact of hail events. The spatial coverage of a hailstorm, which can be measured by hail size reports, remote sensing products, or the extent of insured dam-

ages, is of great importance for assessing the hail risk of an area. Analyzing the environmental characteristics associated with hailstorms has the potential to advance our understanding of hailstorm processes, microphysics, and prediction. By examining these factors during hailstorms, we can gain valuable insights into the dynamics and mechanisms at play, contributing to the broader knowledge in this field.



Despite its importance, accurately estimating the size of hail or the severity of hail damage remains a challenge. Currently,
there are three main approaches for estimating hail severity: hail measurements at ground level, insurance claims data, and
weather radar data. Direct observations of hail can be segregated into two categories, reports and in-situ measurements. Reports
have biases related to population location, diurnal sampling bias, and size clustering. In-situ measurements like disdrometers
or hail pads, are the most accurate but are generally sparse or deployed across small areas Allen and Tippett (2015).

Insurance data is more widespread than in-situ measurements. However, it has two main limitations. First, it is restricted to
developed or populated areas where insured properties exist. Second, it only provides the cost of damages, which is highly de-
pendent on the value of the property, how vulnerable is the property to hail damage, low-level winds, the Hail Size Distribution
(HSD), and the density of hailstones (Giammanco et al., 2015).

Radar-derived hail products have the advantage of high spatiotemporal resolution, increased homogeneity, and coverage,
but two challenges remain for accurate estimation of hail. First, the size distribution and concentration of hailstones cannot
be derived from radar reflectivity alone. Reflectivity is the sum of contributions from individual hydrometeors, therefore is
highly dependent on both its size and concentration (Dennis and Kumjian, 2017b). Similarly, a mixture of liquid and frozen
hydrometeors could be present in the volume and result in a size estimate with a positive bias. Second, hailstones are advected
by environmental and storm-generated winds during their descent, which can result in a mismatch between the radar-estimated
hail location from observations aloft and ground observations. The use of polarimetric radars can improve the quality of hail
size estimates from radar by providing additional observations related to the size, shape, and orientation of the hydromete-
ors (Depue et al., 2007; Kumjian and Ryzhkov, 2008; Ortega et al., 2016). However, long term polarimetric radar observations
required to create a hail climatology are lacking in most locations; and some areas lack polarimetric radar coverage. The sec-
ond limitation can be mitigated by modeling the trajectories of the hailstones (Brook et al., 2021), assuming three dimensional
wind information is available; using only low-level information for hail estimation (Depue et al., 2007; Ortega et al., 2016);
or by matching hail size reports or insurance claims to radar-derived hail products within a defined spatiotemporal radius of
influence (Cintineo et al., 2012; Nanni et al., 2000; Warren et al., 2020). However, these mitigation strategies have limitations:
3D wind observations are unavailable in most locations, low-level information is only available close to the radar, which re-
duces coverage and is more prone to data quality issues such as ground clutter and beam blockage. Additionally, products that
estimate hail from reflectivity above the freezing level are often not representative of conditions near the ground. Moreover,
this matching requires a sufficiently large sample of hail size reports or insurance claims.

Despite the limitations of radar-derived hail products, they remain the most effective tool for estimating hail occurrence,
calculating hail risk climatologies, and providing situational awareness for operational forecasters. For example, the Aus-
tralian Bureau of Meteorology (BoM) uses the Maximum Expected Size of Hail (MESH) as guidance for issuing thunderstorm
warnings (Richter and Deslandes, 2007). MESH was originally calculated by fitting the Severe Hail Index (SHI) to the 75[th]
percentile of 107 maximum hail size reports using a power-law function (Witt et al., 1998). This was later improved by using a
larger number of reports, 5897, and fitting to the 75[th] and 95[th] percentiles (Murillo and Homeyer, 2019). The SHI is a weighted
vertical integration of hail kinetic energy above the melting layer, which is estimated from radar reflectivity (Witt et al., 1998).



In our study, we leverage MESH data from the BoM's national network of weather radars, combined with a 10-year dataset of hail damage insurance claims provided by Suncorp Group Limited (Suncorp) and meteorological data from ERA5 reanalysis (Hersbach et al., 2020), to train a deep neural network capable of predicting hail damage. The manuscript describes the insurance data and applied post-processing (section 2); the radar data and calculated products (section 3); the procedure to match the insurance data to radar observations (section 4); the development and evaluation of the neural network (section 5); and finally the relationship between predicted hail damage, MESH, and meteorology (section 6).

## 2   Insurance Data

Building-scale insurance data were provided by Suncorp for all of Australia, which included location, event occurred date, sum insured, and incurred loss. The dataset also provided information on the insured property's characteristics. Data were limited to areas covered by radar observation and encompassed the time period January 2010 - June 2022, which resulted in 311,196 individual damage claims. To prevent any biases towards more expensive properties, we calculated a damage metric as the ratio of incurred loss to insured sum also known as loss ratio, hereafter referred to as simply damage). It is important to note that our study only investigates hail damage on buildings, while damage on vehicles was not provided due to uncertainty with the damage location.

### Archetype normalization

We recognize that various property types can exhibit different levels of loss ratio from the same hail size and concentration (Blong, 2007; Brown et al., 2015; Hohl et al., 2002; Mobasher et al., 2022). For example, certain roof types are more susceptible to hail damage than others, and the presence of tree coverage can also influence observed hail damage. Accurate analysis and comparison to the radar estimates requires minimizing the influence of these variables on the observed damage. To achieve this goal, we identified all possible combinations of property characteristics (roof type, wall type, construction year, tree coverage) and selected the most frequently occurring combinations (archetypes), such that at least 85% of policies are represented. This resulted in 12 archetypes with varying levels of vulnerability, 11 archetypes accounted for more than 85% of the policies, with the rest labelled as other (the 12th archetype). The mean damage for each archetype and for the full data was calculated (Figure 1). The results indicated that some archetypes were over three times more vulnerable to hail damage than others. To ensure the overall archetype mean equalled the unscaled mean of the full data, the damage for each archetype was rescaled to match the overall mean (Figure 1, top-right). Note that the archetype details are commercially sensitive and are not shown in this study.

### Event identification

Insurance data were limited to hail damage claims but the percentage of days with claims in most radar domains did not accurately represent the true hail frequency, with most days recording at least one claim. The majority of these days recorded less than ten claims each, while a small number of days saw claim spikes in the thousands (Figure 2). To address this issue,





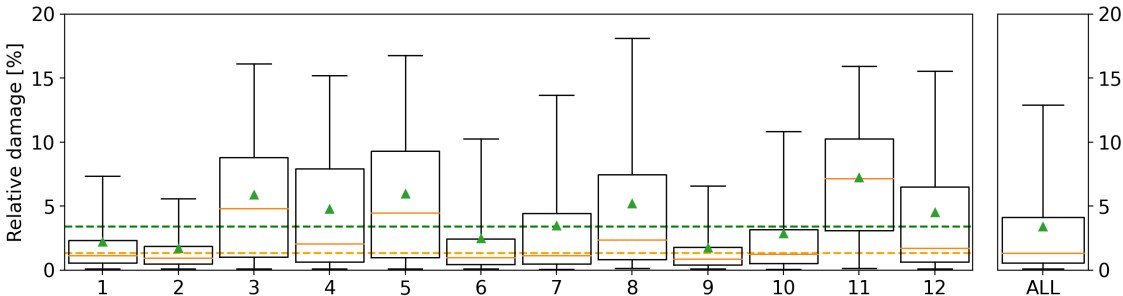

**Figure 1.** Relative damage distribution for each archetype. The black box shows the extent of the 25[th] and 75[th] percentiles of the data; the orange line shows the median, and the green triangle the mean. The whiskers show the 5[th] and 95[th] percentiles. The dashed green and orange lines show the mean and median of the full dataset, respectively.

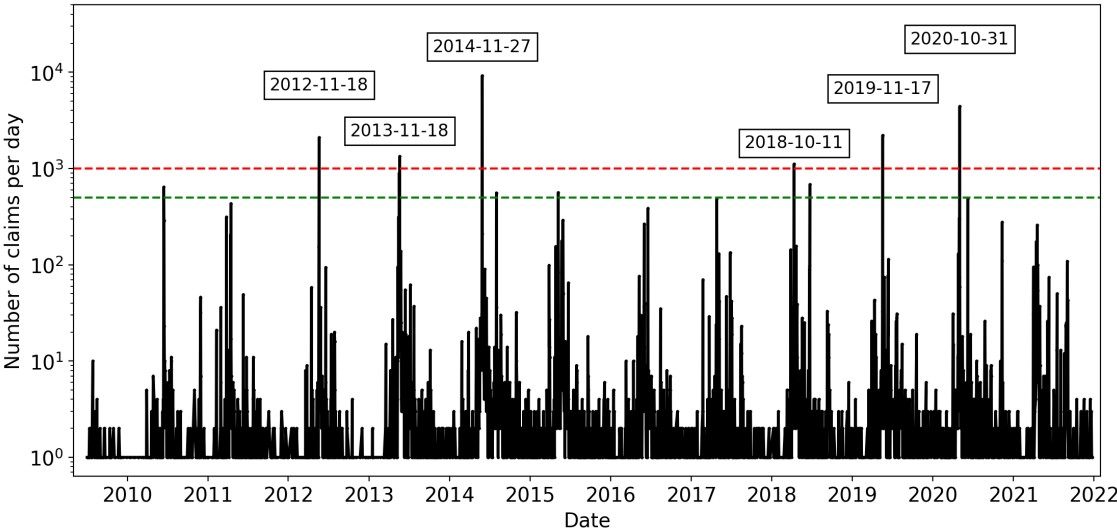

**Figure 2.** Time series of daily claims for the Brisbane radar domain (Mt Stapylton). Events with more than 999 claims are labelled. A maximum distance of 150 km to the radar was used. The green and red dashed lines show the 500 and 1000 thresholds, respectively.

we defined intense hail events as days with at least 1000 claims and days with between 500 and 999 claims were classified as medium hail events. A close look at the days surrounding these events show increased claim counts above the baseline starting around one week before and returning to baseline about one week after the peak, potentially due to mislabeling of the date or errors in the reported date of damage. To mitigate this, claims withing $\pm$ 7 days from each peak where included in the event's dataset.





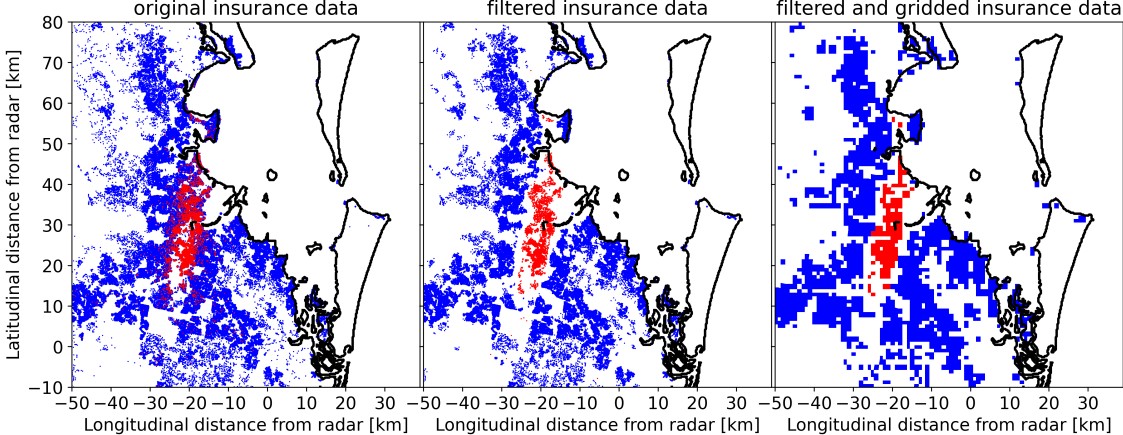

**Figure 3.** Effect of filtering insurance data; red points show valid policies with reported damage, blue dots show valid policies with no reported damage and white spaces show no valid policies. Left panel shows the raw data, middle panel shows only policies after filtering, and the right panel shows the gridded filtered data for the event.

**Claims filtering**

To ensure that only regions with sufficient exposure and a substantial amount of damaged properties were included in the analysis, the insurance dataset underwent filtering. To remove areas with insufficient exposure, policies were considered only if they were valid during an event and at least 10 other policies were valid within a 1 km radius surrounding the policy (policies within a circle of 1 km radius centered at the policy to be evaluated). To identify areas of substantial damage, damage claims were only considered if at least 5% of the policies within a 1 km radius also reported damage, otherwise the claims were

removed. To identify areas of no damage, if a policy did not report damage but more than 1% of neighboring policies (within a 1 km radius) reported damage, the policy was removed. The difference between the two percentages ($\geq 5\%$ for damage areas and $\leq 1\%$ for undamaged areas) creates a "buffer zone" where the occurrence of hail damage was uncertain. Once the above filtering was done, the insurance data was gridded such that it exactly match the radar's grid for each domain. This was done by calculating the mean damage within a borders of each radar's grid box. Note that this results in grid points with mean damage

above zero that is only representative of the damage claims, as undamaged policies would be removed from such ares. These filtering thresholds were identified empirically and can be varied by 10% without noteworthy effect on the analysis. Figure 3 illustrates the effect of this filtering and the subsequent gridding. Applying all aforementioned filters resulted in 18 intense hail events, 12 medium hail events, 1,557 damage grid points, and 45,209 exposed but undamaged grid points. On average, the filtering removed 21.4% of damage claims and 12.3% of exposed but undamaged policies.

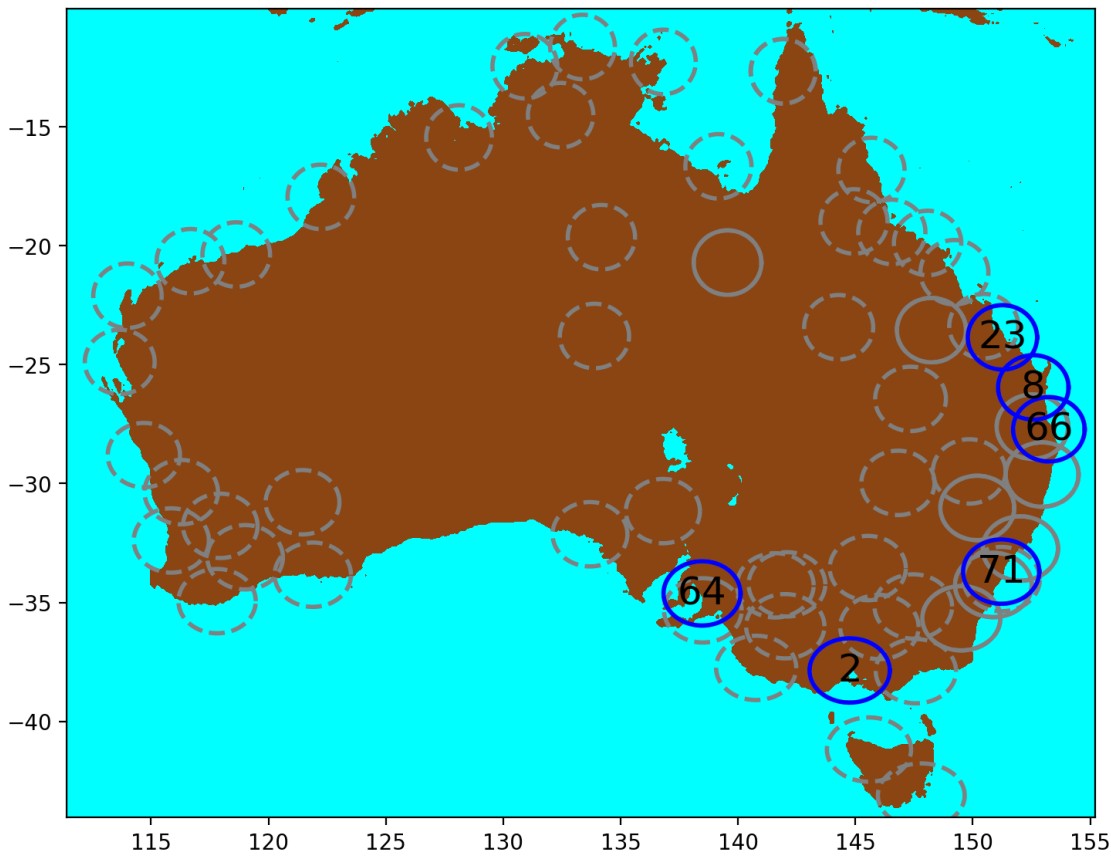

**Figure 4.** Map of the full Bureau's radar network. The circles show the coverage of each radar and the number refers to the radar ID found in Table 1. S-band radars are shown with solid circles. Blue circles show the radars used in association with insurance claims.

## 3 Radar Data

In this study, data from the Bureau's radar network was utilized (Soderholm et al., 2022), with a focus on selecting S-band radars that are better suited for hail observations compared to C-band radars (Ryzhkov et al., 2013). Although this led to a reduction in the number of radar sites used in the analysis, it still covered a large proportion of the population and claims. The geographical distribution of these radars can be observed in Figure 4 and configuration details can be found in Table 1. The radar reflectivity was calibrated following the method outlined in Louf et al. (2019), and gridded at a 1 km horizontal and 500 m vertical resolution utilizing the methodology described in Dahl et al. 2019, appendix A, was implemented, which uses linear interpolation in elevation, and radius of influence on the azimuth-range space. Grid points too far ($> 150$ km) or too close ($<$ 6 km) to the radar site were excluded from the dataset.



**Table 1.** Australian S-band radar network details.

| Radar ID | City | Radar type | Start date | Latitude | Longitude | Beamwidth |
|---|---|---|---|---|---|---|
| 2 | Melbourne | Meteor1500SDP | 07/09/1993 | -37.8553 | 144.7555 | 1 |
| 3 | Wollongong | DWSR8502S | 22/10/1995 | -34.2624 | 150.8751 | 1.9 |
| 4 | Newcastle | DWSR74S | 07/09/1999 | -32.7298 | 152.0254 | 1.9 |
| 8 | Gympie | DWSR8502S | 08/11/1999 | -25.9574 | 152.5768 | 1.9 |
| 23 | Gladstone | WSR74S | 24/12/1998 | -23.855 | 151.2626 | 1.9 |
| 28 | Grafton | WSR74S | 23/09/1998 | -29.622 | 152.951 | 1.9 |
| 40 | Canberra | DWSR74S | 22/11/2002 | -35.6614 | 149.5122 | 1.9 |
| 50 | Brisbane | WSR74S | 04/11/1994 | -27.608 | 152.539 | 1.9 |
| 64 | Adelaide | Meteor1500SDP | 27/10/2005 | -34.6169 | 138.4689 | 1 |
| 66 | Brisbane | Meteor1500SDP | 08/06/2006 | -27.7178 | 153.24 | 1 |
| 69 | Namoi | DWSR8502S | 02/06/2010 | -31.0236 | 150.1917 | 1.9 |
| 71 | Sydney | Meteor1500SDP | 15/05/2009 | -33.7008 | 151.2094 | 1 |
| 72 | Emerald | DWSR8502S | 09/03/2010 | -23.5498 | 148.2392 | 1.9 |
| 75 | Mount Isa | DWSR8502S | 14/09/2012 | -20.7112 | 139.5552 | 1.9 |

**Severe Hail Index (SHI)**

The SHI quantifies hail severity using a weighted vertical integration of reflectivities above the environmental freezing level (Witt et al., 1998). The resulting output is a 2D gridded map which is indicative of the severity of the hail event. The height of the 0° C and -20° C dry bulb levels were retrieved from ERA5 (Hersbach et al., 2020) from the grid point closest in time to the observation and location to the radar location.

**Maximum Estimated Size of Hail (MESH)**

The Maximum Estimated Size of Hail (MESH) is a quantitative tool that transforms SHI into hail size by fitting SHI to a chosen percentile of maximum observed hail size by using a power curve originally developed by Witt et al. (1998) and improved by Murillo and Homeyer (2019) with a larger report dataset. For this study we use the 75[th] fit for Murillo and Homeyer (2019) dataset. This transformation enables the estimation of hail size from the SHI data.

**Calculation of event SHI swath**

The event SHI was computed for each grid point within a radar's coverage for the event's day $\pm$ one day to account for possible discrepancies between the reporting time of damage. This computation enabled the production of a storm swath map for each event day (Figure 5). The volume scan period for radars used in this study ranges from 5 to 10 minutes, leading to discontinuous





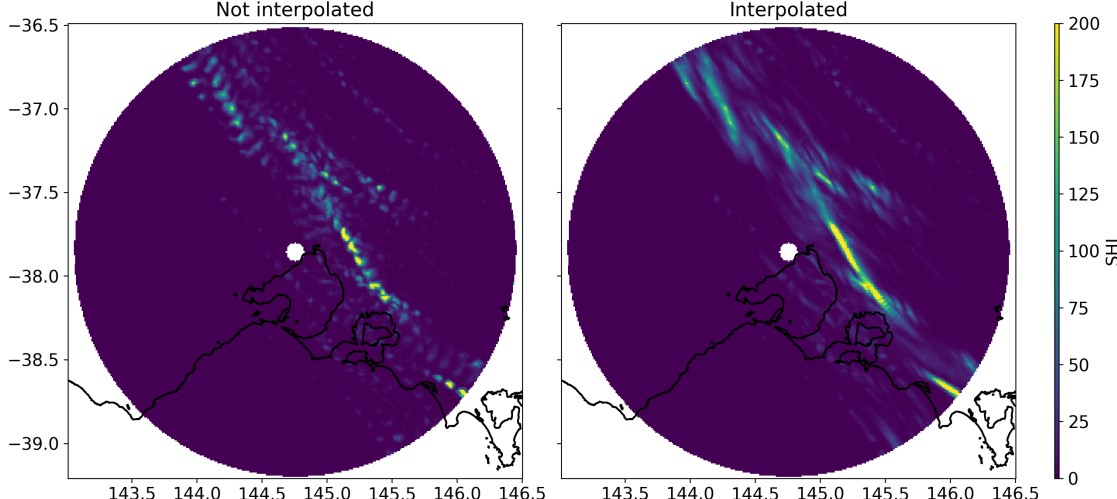

**Figure 5.** Severe Hail Index (SHI) daily maximum for a sample event. The left panel shows a discontinuous storm track due to the time gap between successive radar scans. The right panel shows the same day with interpolated data based on the optical flow of SHI.

SHI swaths, which are most pronounced for fast-moving storms (Figure 5, left panel). To mitigate this spurious discontinuity, an interpolation algorithm was employed, which uses the estimated field advection from its optical flow to fill in these gaps

(Figure 5, right panel). This is done using the scipy python package, specifically the minimize method of the optimize class which applies the Nelder-Mead minimization algorithm (Nelder and Mead, 1965). This algorithm compares two subsequent scans (images) and attempts to minimize the difference between the two images by displacing one and comparing to the other. Once the optimal displacement is computed, a linear interpolation between the two time stamps is calculated. Once the interpolation between scans is computed, the maximum SHI for each grid point is retained from the two original time stamps

and the interpolation, along with its corresponding timestamp, allowing for the retrieval of the associated meteorological conditions from ERA5.

**Associated meteorological data**

The calculation of SHI required information on the 0° C and -20° C dry bulb levels, which is derived from the ERA5 pressure level dataset for each radar location. In addition to these variables, other meteorological variables were extracted from ERA5 at

ground level and freezing height, these were: 3D wind components, specific humidity, divergence, vorticity, and atmospheric pressure. CAPE and CIN (most unstable Convective Available Potential Energy and Convective Inhibition, respectively), were also extracted; for more details on the calculation of these variables see Groenemeijer et al. 2019 All these variables were retrieved for each grid point at the time when the maximum SHI of the day occurred. These data served as the input for training the neural network in addition to SHI and the observed damage.



## 4   Claims and Radar Matching – Virtual Advection

Hail retrievals from radar observations aloft often do not align with observations at ground level, resulting in a common mismatch (Brook et al., 2021). MESH/SHI are calculated from reflectivities above the environmental freezing level, which can reside several kilometers above the surface. Depending on the strength of storm-generated and environmental winds, hail descending below this level may be advected several kilometers from is initial location, leading to the aforementioned mismatch. By design, MESH (and, by extension, SHI) would have greater skill as a predictor of hail damage during events where hail falls with little horizontal movement. To address this issue, we developed a Virtual Advection (VA) algorithm, which matches ground-level observations of hail damage to the appropriate radar observations aloft, mitigating this error. Similar approaches have produced substantial improvement of the correlation between observed damage and radar derived hail estimates (Hohl et al., 2002; Schiesser, 1990; Schmid et al., 1992; Schuster et al., 2006)

**VA Algorithm and assumptions**

To apply the VA algorithm, we made certain assumptions. We assumed that the highest damage within a local area corresponded to the highest observed SHI values aloft. However, this assumption may not hold true in cases where the hailstorm area is not densely covered by insured properties, and large hailstones may fall in uninsured areas or areas without buildings. In such cases, this assumption would systematically lead to a positive bias in SHI values. To address this issue, we filtered the claims data to only include grids with sufficient exposure, as described in Section 2.

To match damage to SHI, we first sorted each events' normalised gridded damage dataset in descending order according to value. Then, we matched the first (highest) damage grid point to the highest SHI grid point within a 4 km Radius-of-Influence (RoI) around the damage grid point. Once we established a pair of damage and SHI grid points, we stored the pair in a new VA dataset with its associated horizontal displacement vector and meteorological variables (from ERA5). We removed this SHI grid point from the available SHI grid and repeated the process until we matched all damage grid points to SHI observations. Once there are no more damage grids, the average horizontal displacement vector is calculated. The matching is restarted but with the average horizontal displacement vector already applied to the 4 km RoI. This double pass approach allows to correct for the environmental wind displacement (represented by the average horizontal displacement vector of the first run), and the storm generated winds (done in the second run). The 4 km RoI was selected empirically, with higher values showing only small improvement in the relationship between SHI and damage, and smaller numbers showing poor visual matching of the radar and damage swaths. When both runs are used, up to 8 km of displacement can be achieved, as both runs could result in displacements in the same direction for a given claim/SHI grid pair. However, this was rare for the studied events, with most events having an average final displacement (after the second run) less than 2 km, and none of the events having average final displacements above 4 km. Note that no local consistency in the displacement vector is not enforced for the second run, which allows convergence and divergence of the SHI field. If all non-zero SHI observations are used before matching all damage grid points, then the closest zero SHI observation is used within the RoI. For grid points with contracts and zero observed damage,



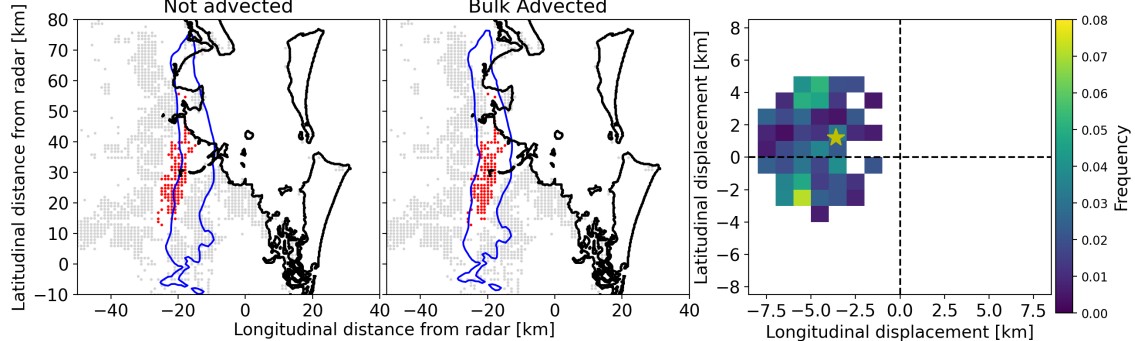

**Figure 6.** Sample of virtual advection of severe hail event (Brisbane 2014). Left panel shows grid points with exposure that reported no damage in grey, damaged grid points in red, and grid points with insufficient exposure in white. The blue contour line is derived from MESH at 40 mm. The middle panel shows the same but with MESH displaced by the weighted average (by damage) of the virtually advected data. Right panel shows the histogram of the horizontal displacement of MESH to match the damage at ground level, the yellow star shows the weighted average. The dashed range ring is 150 km in radius.

we matched the lowest SHI observations (including zero) within the RoI instead of the highest. Grid points with no valid SHI observations within the RoI due to being too far or too close to the radar site were excluded from the VA dataset.

We assessed the performance of the VA algorithm using the 2014 Brisbane hailstorm, a well-known event with substantial
advection that resulted in a large mismatch between MESH (SHI) observations and reported damage locations (Brook et al., 2021; Warren et al., 2020). The left panel of Figure 6 shows this mismatch. To visualize the algorithm's performance, we calculated the weighted average (by damage) of the horizontal VA displacement vectors (yellow star in the right panel of Figure 6). We then displaced the original MESH grid by this average vector (middle panel of Figure 6), resulting in much better agreement with the observed damage. We refer to this displacement by the average vector as bulk advected, to differentiate
it from the data produced by the VA algorithm, which is visualized in the 2D histogram of Figure 6 (right panel). When comparing this bulk advection with Brook et al. (2022) individual modeling of the hailstone trajectories for this event, good agreement can be observed for the overall swath displacement. In addition, Brook et al. (2022) found that the average motion vector for this event was 2.1 km in a northwesterly cardinal direction, which is very similar to our aforementioned weighted average horizontal displacement (2D histogram in Figure 6).

**Performance comparison**

The VA algorithm was applied to all hail events, and the resulting MESH–damage dataset is presented in the left panel of Figure 7. The right panel of this figure shows the performance diagram (Roebber, 2009) for various MESH thresholds above which damage is predicted for each advection correction type. Here the 40 mm threshold shows the highest Critical Success Index (CSI) for all correction types; this threshold is shown on the left panel (green dashed line), revealing that most damage
occurs for MESH values above this threshold. This diagram illustrates clear improvement for the advection-corrected sets, with





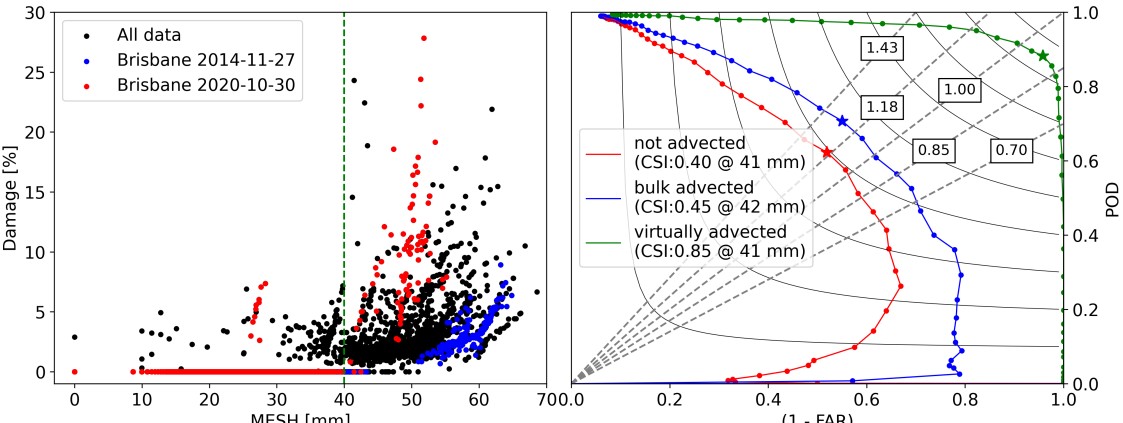

**Figure 7.** Relationship between MESH and damage. The left panel shows virtually advected MESH against observed damage, with the green dashed line showing the best CSI threshold (for all correction types). The right panel shows the Roebber performance diagram for non-advected MESH, bulk-advected MESH, and individually (virtually) advected MESH. The stars in the right panel show the best CSI for each dataset. The grey dashed lines show the bias with their respective values in boxes.

the VA algorithm substantially outperforming the bulk advection method. The scatter plot in this panel highlights two events in blue and red, that exhibit distinct MESH–damage relationships. Note that since the contours of individually advected MESH are not well defined for non-damaging hail and the relationship between the radar hail estimate and observed damage is poor for the bulk-advected dataset, maps of radar derived hail products for all following sections are bulk-advected; while scatter
plots comparing these products to damage are individually advected.

## 5   Hail Damage Estimate (HDE)

In this section, we explore the limitations of using MESH as a predictor of hail damage intensity and present a novel approach for estimating hail damage. While the VA algorithm improves the performance of MESH as a hail damage predictor when above 40 mm, its ability to estimate damage intensity remains inconsistent across different events, as demonstrated in the left panel
of Figure 7. This inconsistency suggests that the relationship between MESH and hail damage intensity is dependent on other factors specific to each event, such as the meteorological condition conducive for hail growth. Furthermore, radar reflectivity has inherent limitations since radar reflectivity is the integral of the particle size distribution times the diameter to the $6^{th}$ power under the Rayleigh approximation, the same reflectivity could be produced by either a few large hailstones or a high concentration of smaller hailstones, which would result in very different severity of hail damage for the same reflectivity (and
therefore SHI). In addition, the presence of other hydrometeors like ice crystals or liquid water can further decouple observed reflectivity from severity of hail damage. To explore the information content of meteorological conditions for improving hail damage estimate, we developed an artificial neural network that incorporates meteorological variables associated with each



**Table 2.** Variables used as input for the Hail Damage Estimate (HDE) and their relative importance.

| *Neural network input variables* | *Description* | *Mean relative input importance and standard deviation*[%] |
|---|---|---|
| SHI | Severe Hail Index | $35 \pm 5$ |
| Q_700hPa | Specific humidity at 700 hPa pressure level | $15 \pm 3$ |
| WS_0C | Wind Speed at 0 C dry bulb level | $15 \pm 4$ |
| V_0M | Meridional wind component at ground level | $13 \pm 5$ |
| CAPE | most unstable Convective Available Potential Energy | $12 \pm 4$ |
| CIN | most unstable Convective Inhibition | $10 \pm 3$ |

grid point to improve the SHI–damage relationship. Our approach generates Hail Damage Estimates (HDE) that demonstrate better agreement with the observed damage.

**Neural network structure and selection of meteorological data**

To develop and train the HDE neural network, we utilized TensorFlow and Keras (Abadi et al., 2015).

Initially, a wide range of meteorological variables were incorporated into the model. We then applied the Shapley Additive Explanations (SHAP, Lundberg and Lee 2017) analysis to identify the most skillful variables, i.e., those with the greatest impact on the prediction (Table 2). This allowed us to maintain the model's accuracy while minimizing its complexity. While 225 initially the network was deeper, it was optimized to maintain accuracy while minimizing computational cost; our final configuration consisted of six layers, each containing 6 (input), 9, 7, 6, 3, and 1 (output) neurons, all of which were densely connected and activated using a rectified linear activation function (excluding the output neuron, which was linear). We linearly normalized the input variables to ensure their values ranged between 0 and 1.

**Training and performance**

To train our model, we utilized the VA dataset, which contained only the 18 intense hail events (i.e., those with over a thousand claims in a single day). We had access to 46,766 distinct damage-SHI points that matched our criteria, with 18,899 points indicating no reported damage and zero SHI, 26,132 indicating no damage but SHI > zero, and 1,556 indicating both reported damage and SHI above zero. Due to the highly unbalanced nature of the data (i.e., no damages greatly outnumber damages), we set the model's initial bias to the natural logarithm of the ratio between the damage count and the no damage count (-3.365). By 235 setting the initial bias to the logarithm of the class ratio, we are effectively providing the model with a starting point that takes into account the class imbalance (He and Garcia, 2009). We randomly separated the data into two groups: a training dataset (80%) and a validation dataset (20%). The meteorological data associated with non-zero SHI points is well defined, as the time when this SHI occurred is known. This is not the case when a grid point with zero SHI (i.e., outside the storm swath) and any





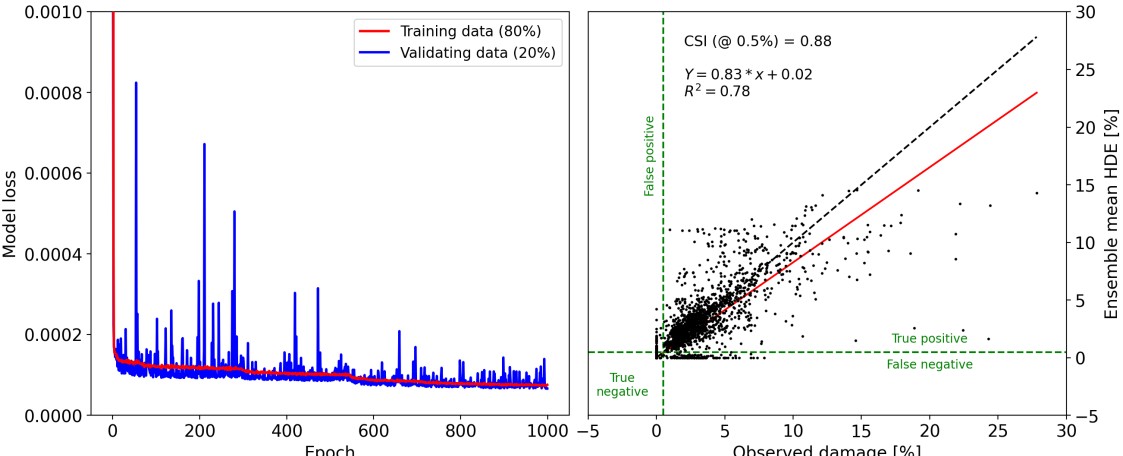

**Figure 8.** Left panel shows a sample training time series of the model's loss for the training and validation data. Right panel shows the relationship between observed damage and HDE's ensemble mean. CSI is calculated based on 0.5% damage threshold (green dashed lines). The red line shows the best fit with its equation and coefficient of determination shown on the top left corner. The dashed black line shows the 1:1 relationship.

time and therefore associated meteorological data can be assigned. To ensure that the meteorological data used for training
was representative of both hail and non-hail atmospheric conditions, we selected random time stamps within $\pm 1$ day of the event for grid points where the SHI was zero. We trained the model 650 times for 1000 epochs each, with random separation of the two sets (training and validation) and randomly initialized model weights but the same initial bias. Figure 8 (left panel) displays a representative training time series, demonstrating that the trained model performs equally well on the training and validation datasets. To calculate HDE, we utilized an ensemble approach with five members that maximized CSI and coefficient
of determination ($R^2$). This number of members showed the best balance between computational cost and performance, with additional members showing minimal improvement on the ensemble's skill. The resulting ensemble mean yielded a CSI of 0.88 and an $R^2$ of 0.78 compared to observations for the full dataset, including the 12 medium events which were completely absent from the training. However, the model tends to underestimate large damages ($> 10\%$), as depicted in the right panel of Figure 8, this is likely due to the under-representation of such cases in the dataset.

**HDE - MESH relationship**

To understand the relationship between HDE and the conventional MESH retrieval, a much larger HDE dataset was required than that available from VA dataset. Therefore HDE and MESH were calculated for all S-band radars of the Australian operational weather radar archive. In Figure 9, we present the data obtained. A sigmoid function (equation 1) provided the best fit for the data while representing a physically realistic relationship, where damage asymptotically approaches 100% as MESH
tends to infinity.

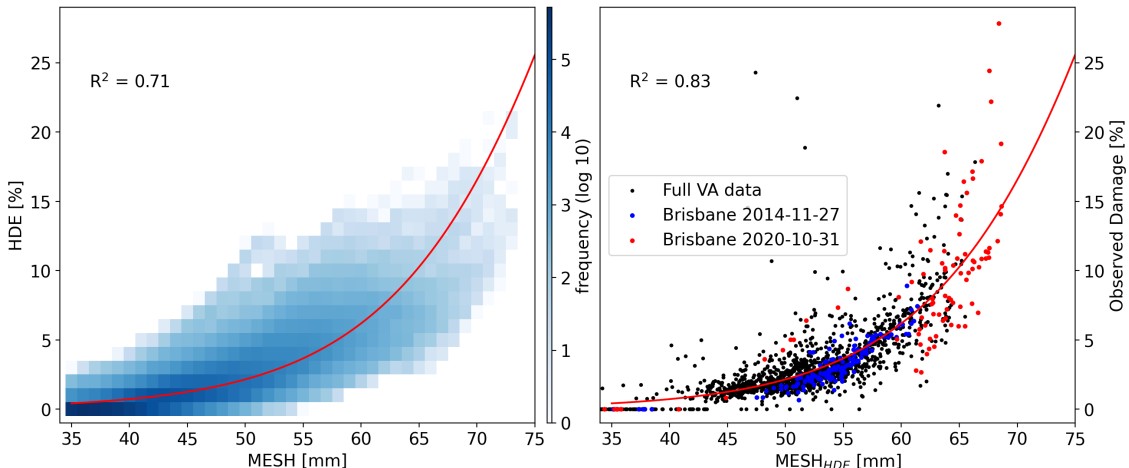

**Figure 9.** Left panel shows HDE and MESH calculated for the full radar archive as a 2D histogram with 1% and 1mm bins with the best fit of equation 1. Right panel shows the relationship between observed damage and MESH$_{\mathrm{HDE}}$, with the previously highlighted events shown in red and blue.

$$HDE = \frac{100}{1 + e^{-A(MESH-B)}} \tag{1}$$

This type of relationship has been used before to relate radar-derived hail estimates to hail damage (Schiesser, 1990). Only points where MESH > 35 mm are shown, as the remaining data points were too numerous to display and damage tends to zero. The fit is relatively good (R$^2$=0.71), indicating a reasonable correlation between MESH and HDE. Using the inverse of equation 1, we calculated MESH$_{\mathrm{HDE}}$, which allows for an investigation of how the neural network adjusts the radar observations (SHI) using the environmental information. Note that MESH$_{\mathrm{HDE}}$ is not intended to replace MESH as a reliable size of hail, but rather as a tool to identify where the model has decreased or increase hail intensity according to the environmental conditions.

We then examined the relationship between MESH$_{\mathrm{HDE}}$ and observed claims for the events captured by the damage dataset, as shown in the right panel of Figure 9, with the two Brisbane events previously highlighted in the same colors. The relationship is greatly improved, and both events now show more consistent sizes relative to the observed damage.

**HDE case studies**

In this section we show the performance and behavior of HDE for the two previously highlighted hail events to analyse how different meteorological conditions drive HDE behaviour.



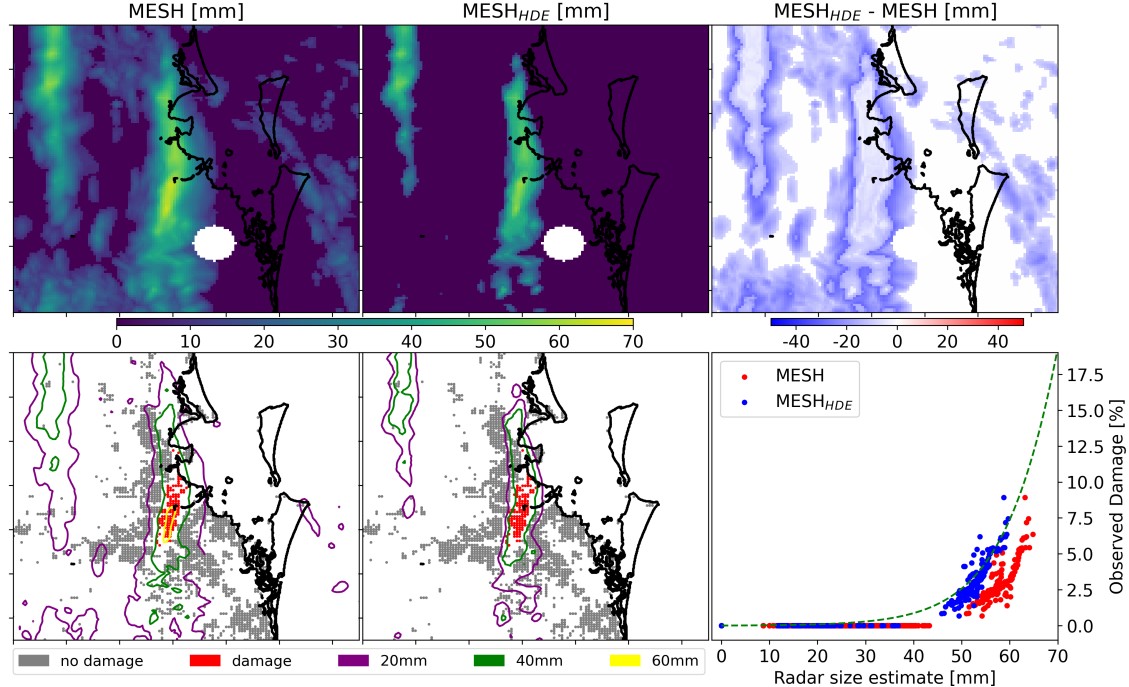

**Figure 10.** Comparison of MESH, MESH$_{HDE}$, and observed damages for an intense hail event in Brisbane on 2014-11-27. Top row shows (from left to right) maps of MESH, MESH$_{HDE}$ (MESH derived from HDE), and the difference between the two. Bottom left and middle show observed damage with bulk-advected MESH and MESH$_{HDE}$ contours overlaid. Bottom right panel shows the relationship between observed damage, and the grid-point advected MESH and MESH$_{HDE}$.

### Brisbane 2014 hail event

On November 27$^{th}$, 2014, Brisbane experienced a severe hailstorm that caused substantial damage to a densely populated area, resulting in over AUD 1.5 billion losses (normalized to 2017) and more than nine thousand individual building claims (for Suncorp). Giant hailstones, reportedly measuring around 70 mm in size, were observed during this event (Parackal et al., 2015). In Figure 10, we present MESH and MESH$_{HDE}$ for this event. We observed that MESH$_{HDE}$ assigns low values of MESH to zero, due to low MESH values inability to cause any damage to property and therefore resulting in zero HDE. Additionally, within the eastern-most storm cell, where MESH is high and damage occurred, MESH$_{HDE}$ was mostly lower than MESH, as highlighted in the bottom-right panel. This finding shows that MESH$_{HDE}$ more closely aligns with the mean fit (green dashed line in the bottom-right panel) and corrects the observed positive bias of this event when compared with the original VA data. While MESH$_{HDE}$ shows a more consistent fit, it is evident that it is not a good representation of the actual size of expected hail as it is still far below the observed size for this storm. The bottom left and middle panels show the MESH and MESH$_{HDE}$ contours over the observed damage respectively.



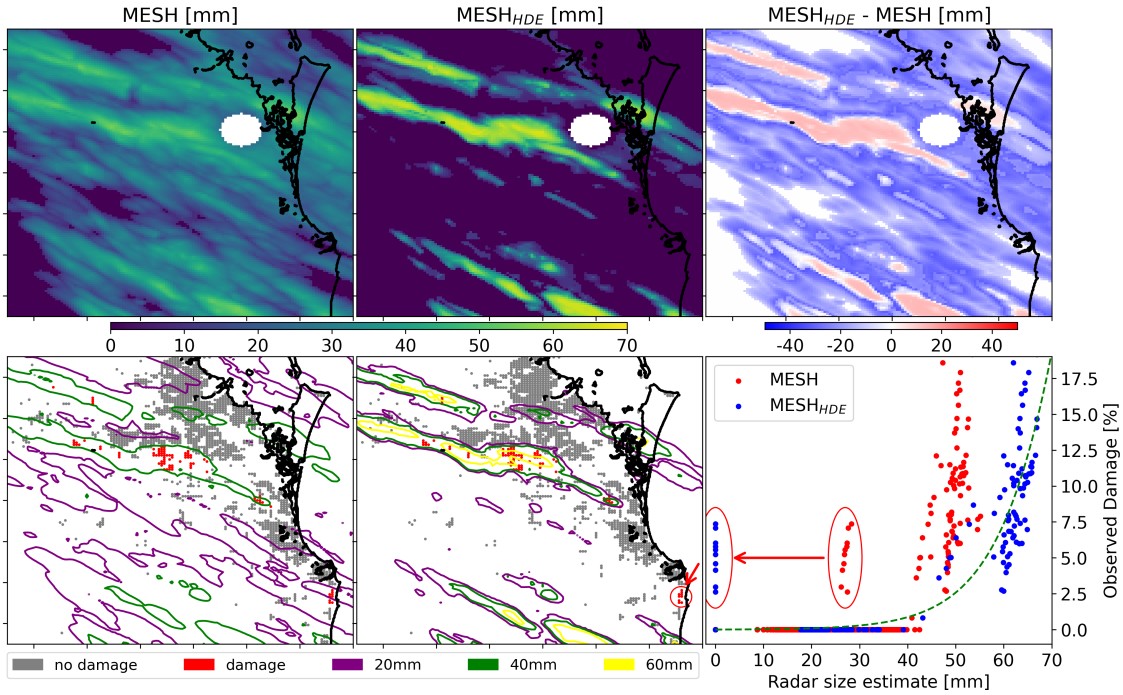

**Figure 11.** Same as Figure 10 but for an intense hail event in Brisbane during 2020-10-31. The red circles show potential spurious claims.

**Brisbane 2020 hail event**

In contrast to the 2014 event, the 2020 Brisbane hail event showed a negative bias in MESH relative to the full VA dataset. As shown in Figure 11, $MESH_{HDE}$ was lower than MESH for most of the domain, but considerably higher than MESH where hail damage occurred. It is worth noting that this case exhibited some potentially spurious claims, as highlighted within the red circles in the bottom-right panel. These damage grid points were relatively far from the main storm swaths (indicated with red arrows in the lower middle subplots) and could be due to the misclassification of damage cause (i.e., wind damage instead of hail) in the insurance dataset.

## 6   Hail damage, MESH, and meteorology

The 2020 event was associated with very different meteorological conditions in the ERA5 reanalysis than the 2014 event, with higher CAPE and lower CIN, stronger northerly winds at ground level, lower humidity aloft, and increased winds at the melting layer (Table 3). Looking at the MESH values from these events alone, one would expect the 2014 event to have caused more damage than the 2020 event, as the mean MESH values in the storm cores (here defined as the areas with HDE above zero) were 58.3 mm and 51.6 mm, respectively. Nevertheless, in the observed damage dataset, the 2014 event produced lower





**Table 3.** Mean values of MESH, $MESH_{HDE}$, and environmental parameters for storm cores during the 2014 and 2020 Brisbane events. The error bounds show the standard deviation within the storm cores.

| Variables name | 2014 event mean | 2020 event mean |
|---|---|---|
| MESH [mm] | $58.3 \pm 3.1$ | $51.6 \pm 1.3$ |
| $MESH_{HDE}$ [mm] | $51.3 \pm 4.5$ | $61.4 \pm 5.1$ |
| Q_700hPa [$kg \cdot kg^{-1}$] | $0.0056 \pm 0.0001$ | $0.0031 \pm 0.0002$ |
| CAPE [$J \cdot kg^{-1}$] | $526.9 \pm 0.5$ | $3137.8 \pm 21.8$ |
| CIN [$J \cdot kg^{-1}$] | $282.3 \pm 49.3$ | $88.7 \pm 28.7$ |
| WS_0C [$m \cdot s^{-1}$] | $7.20 \pm 0.07$ | $27.16 \pm 0.98$ |
| V_0M [$m \cdot s^{-1}$] | $1.72 \pm 0.48$ | $-10.59 \pm 0.16$ |

damage than the 2020 event, as clearly visible in Figure 7. This discrepancy is not observed in $MESH_{HDE}$, with mean values of 51.3 mm and 61.4 mm, for the 2014 and 2020 events respectively.

The cases mentioned highlight the possibility of storms with relatively low MESH values causing severe damage, such as the one that hit Brisbane in 2020, or relatively high MESH values leading to comparatively lower damage, as seen in the 2014 Brisbane storm. The disparity could be attributed to a mixture of hail and other hydrometeor types (liquid water, ice

crystals) which would result in a higher MESH value than a volume containing only the hail component. Further, MESH uses the environmental freezing level, but the updraft's freezing level could be well above the environmental freezing level which could involve larger liquid water droplets than conventional supercooled liquid water droplets (Kumjian and Ryzhkov, 2008). The disparity could also be attributed to differences in the HSD in these volumes. Volumes with different HSD but equivalent scattering response would produce similar MESH values but varying ground damage. Since supercooled liquid water (SLW)

droplets are considerably small, most of the reflectivity signal is expected to be caused by hail, smaller frozen hydrometeors or large liquid water droplets lofted above the environmental melting layer.

Extending this method, we can utilize the difference between $MESH_{HDE}$ and MESH (hereafter $\Delta MESH$) as an indicator of the skewness of the HSD within a storm and/or mix hydrometeor volumes. Positive $\Delta MESH$ indicates a larger proportion of damaging hailstones and/or less proportion of mixed hydrometeors in the volume, while negative $\Delta MESH$ indicates the

contrary.

Figure 12 shows the relationship between each meteorological parameter and HDE and $\Delta MESH$ as box-plots for the full radar archive. Using 40 mm for MESH and 0.5% for HDE as thresholds for damage, and HDE as the truth, the leftmost column represents environments where MESH indicates a false-positive for hail damage while the second column shows opposite environments, where MESH indicates a false-negative. The Kolmogorov-Smirnov (Hodges Jr, 1958) test was applied

to each meteorological variables for the false-positive and false-negative cases, showing that while all sets showed significant differences (p-values approaching zero), specific humidity aloft showed a substantial difference between the two cases, with a D-statistics of 0.64, which is evident from the corresponding box-plots. A clear threshold was identified based on the analysis.





It was found that 82% of the false-positive occurrences happened in environments where the specific humidity at 700 hPa was above 0.0053 kg · kg$^{-1}$. Coincidentally, the same percentage of false-negative occurrences happened in environments where the specific humidity at 700 hPa was below that humidity threshold. By utilizing this threshold, by applying it only for MESH values between 39 and 41 mm, which account for approximately 50% of the false-positive and false-negative cases, the CSI of MESH can be improved from 0.76 (@ 40 mm against HDE) to 0.79. Environments with false-positive results exhibited conditions similar to those in negative $\Delta$MESH bins and low HDE bins. This indicates that most of these false positives resemble environments associated with negative values of $\Delta$MESH and are less likely to have high HDE. False negative cases were associated with low HDE and showed similarities to environments with a positive $\Delta$MESH bias, these cases had an average HDE of 0.8% and a maximum of 4.2%. CAPE tends to increase with $\Delta$MESH, whereas CIN is larger for more extreme $\Delta$MESH of either sign. Looking at the relationship to winds, there is a positive correlation with $\Delta$MESH, except for the extreme positive end. In a similar way these extreme values of $\Delta$MESH are associated with northward winds at ground compared to the rest of the distribution that exhibits no trend.

This analysis indicates that environments with relatively low specific humidity, high CAPE and CIN, low wind speeds aloft and northward winds at ground are likely to have HSD with higher proportion of large hailstones or a lower proportion of other hydrometeors. Regarding environments associated with extreme values of hail damage, these are associated with high CAPE and CIN values, lower wind speeds at the melting layer, and northward winds at ground. No clear signal with respect to specific humidity aloft is observed. A recent modeling study on hail production (Lin and Kumjian, 2022) found that CAPE acts as a modulator to hail growth, with a non-monotonic relationship with hail size which peaks around 2000–2400 J · kg$^{-1}$. Although we do observe that about 50% of the samples with high values of HDE occur close to this CAPE range (Figure 12), these HDE bins also show about 25% of samples with CAPE values above 3000 J · kg$^{-1}$ and a clear positive correlation between CAPE and HDE. Regarding the aforementioned HDE relationship with winds, a similar modeling study (Dennis and Kumjian, 2017a) found that increased deep-layer east-west shear increases hailstone mass while increased low-level north-south shear reduces hailstone mass. Here it is important to note that one of the variables included in the initial input layer of the HDE model was absolute wind speed shear between the surface and the melting layer but was discarded as the SHAP analysis showed that it had low influence on the output relative to the other input variables. The zonal or meridional components of the shear vector were not tested.

## 7 Conclusions

This study has analyzed more than 10 years of insurance and radar data from Australia to investigate the performance of the Maximum Expected Size of Hail (MESH) retrieval in predicting and quantifying hail damage. The results showed that MESH has poor predictive value for damage magnitude estimation, but shows good skill (CSI = 0.88 for MESH larger than 40 mm) as a binary predictor of hail damage when corrected for horizontal advection of hailstones, which is consistent with Brook et al. (2021).



**Figure 12.** Box-plots of the meteorological variables that drive the HDE model. From left to right columns: ($1^{st}$) for HDE < 0.5% and MESH $\geq$ 40mm; ($2^{nd}$) for HDE $\geq$ 0.5% and MESH < 40 mm; ($3^{rd}$) for HDE $\geq$ 0.5% in 2% bins; ($4^{th}$) for HDE $\geq$ 0.5% in 2 mm bins. The $3^{rd}$ and $4^{th}$ columns show the distributiion as a function of HDE and $\Delta$MESH, respectively. Sample size per bins are shown on top. The box-plots follow the same style/structure as figure 1, top panel.



350 To improve damage magnitude estimation, a neural network was trained with meteorological variables from ERA5 in addition to SHI (from which MESH is derived) against observed hail damage. Using SHAP analysis, the most important meteorological variables were identified as specific humidity at 700 hPa, wind speed at the freezing level, northward winds at ground level, CAPE and CIN. This neural network produced a Hail Damage Estimate (HDE) with a high accuracy (CSI = 0.88 and an $R^2$ = 0.78) for estimating hail damage occurrence and intensity.

355 A comparison of HDE with MESH for the full national radar dataset (14 radars with an average of 18.8 years of coverage) revealed a relatively good ($R^2$ = 0.71) fit between the two using a sigmoid curve. This curve was used to derive $MESH_{HDE}$ from HDE with the goal of identifying environments where MESH shows negative or positive biases, potentially due to differences in hail size distributions (HSD) and/or presence of other hydrometeors along with hail.

Environments with negative bias in radar hail estimates are associated with low CAPE, high CIN, and higher specific humidity aloft and are likely to be non-damaging if MESH is below 50 mm. In contrast, environments with positive bias are associated with high CAPE, high CIN, and lower specific humidity aloft and are likely damaging if MESH is above 30 mm. Extreme hail damage was associated with such positive bias environments that in addition showed low wind speeds aloft and northerly winds at ground.

The study provides important insights into the performance of MESH/SHI for estimating property damage and the potential of using neural networks to improve hail damage estimation and identifying patterns between environmental conditions and a storm's HSD and/or presence of mix hydrometeor precipitation. It is important to note that these results were developed for Australian storm environments, and might not be representative of global storm environments, especially given the frequent occurrence of SLW clouds in the region (Ackermann et al., 2021; Chubb et al., 2013). This study was limited to S-band radars, future work should expand this technique to C-band radars. Another limitation to our findings is the relatively small sample of individual storms, which might only sample a subset of all environmental conditions that leads to hail storms. Replicating this work on locations with high population density and radar coverage (i.e., Europe or the USA) would be valuable to potentially mitigate this limitation as well as showing if these same environmental parameters play dominant roles on HDE.

*Data availability.* The insurance data provided by Suncorp is not publicly available for privacy reasons. Details regarding the radar data used in this study can be found in found in the AURA database online at https://www.openradar.io (Soderholm et al., 2022). ERA5 data can be accessed at https://www.ecmwf.int/en/forecasts/dataset/ecmwf-reanalysis-v5 (Hersbach et al., 2020).

*Author contributions.* RW and LY prepared the insurance claims data for analysis and contributed on the interpretation and implementation of the archetype normalization. NR validated and tested the HDE product againts internal Suncorp models. JS prepared the radar data and SHI/MESH hail estimate products. JS and AP contributed in the development of section 4, 5, and the interpretation of section 6. LA designed the paper's outline, carried most of the analysis and wrote the paper. All co-authors participated in paper-related discussions and commented on the paper.





*Competing interests.* The authors have no other competing interest to declare

*Acknowledgements.* This research was funded by Suncorp and the Australian Bureau of Meteorology. We extend our heartfelt gratitude to Dr. Robert Warren for his thoughtful review and insightful comments on the manuscript's draft, his expertise and attention to detail have greatly enhanced the quality and clarity of our work. Coastline data used in all maps were extracted from the Cartopy python package (Met Office, 2010 - 2015).






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
