# Peer review of "Radar and Environment-based Hail Damage Estimates using Machine Learning"

_Atmospheric Measurement Techniques, 2023_

## Author Comment (AC1)

Reviewer 1

General Comments:

This paper presents an analysis of the performance of MESH in predicting insured property hail damage. The authors determined that while MESH alone does not well predict damage magnitude, it has reasonable skill in determining whether damage will occur when corrected for horizontal advection of hail, shown nicely in Figure 6. They developed a neural network that incorporated additional variables from ERA5 and determined which variables were most important in predicting hail damage. They found that the neural network improved the hail damage prediction and identified environments that would lead to positive and negative biases. I found Figures 10 and 11 representing the Brisbane case studies to be particularly helpful in visualizing the calculations and resulting differences between MESH and the improved developed tool.

The techniques presented in this paper advance the state of hail damage prediction knowledge. I commend the authors on their work to relate hail damage data to the meteorological and environmental characteristics. However, there are a few assumptions that need to be stated more clearly. The hail size itself is not the only contributor to damageability. Our paper in ASCE's Natural Hazards Review (Brown-Giammanco, Giammanco, and Estes 2021) showed that the hardness characteristic also plays a role. I realize that it's impractical to gather these data for all events, but this can certainly contribute to explaining why damage may not be perfectly related to the hail size determinations. In addition, the underlying property characteristics (material type, age, maintenance, sheltering) will also play a role in the damageability. Finally, the assumption that the claims data are the "truth" is somewhat problematic, as there will be human judgment biases incorporated. To be clear, I think this dataset is about the best that can be used at this point, so I'm not at all suggesting it's not appropriate—just more pointing out that the limitations of it should be stated.

Thank you for your valuable insights. We have made the necessary revisions to our manuscript in response to your comments. These specific corrections are detailed in the specific comments section provided below, which predominantly address the points you raised.

We have considered the possibility of hail hardness as a contributing factor to the differences between radar observations and hail damage. Consequently, we have incorporated this potential factor into our discussion and analysis.

Specific Comments:

Line 19: It seems as though "measured" may not be the appropriate term to use here. Perhaps "estimated" or "approximated" would be better, since none of the methodologies listed, or the combination of them, really gives a spatial measurement. Related, "or" seems as though it should be "and/or", since you probably increase accuracy by using a combination of methods where possible.

Word "measured" has been changed to "estimated" while "or" has been changed to "and/or".

Line 29-32: I understand that insurance data may be more widely available in Australia, but that is not the case in the United States, and I'm not sure about elsewhere. This is a limitation that should also be noted.

A third limitation was added stating that it might be inaccessible due to policy or privacy concerns.

Line 65: It could be useful to understand the scope of the data, by providing some info on Suncorp. Are they a top-5 carrier in Australia? Or is there an estimate of the percent of properties that they insure? Can you also clarify whether just residential, just commercial, or both kinds of policies were included in the dataset?

Context about Suncorp's market share was added. It was also clarified that only residential policies are included in the dataset.

Line 66: Can you please expand on the property characteristics by providing some examples? Did you get info on age, size, number of stories, any roofing or siding materials used? It looks like maybe the information contained in line 77 is all of the property characteristics? If so, I think it would be good to include at least some examples earlier in line 66.

Done, the provided characteristics were listed and an hypothetical example was provided.

Line 100-101: This choice could introduce some unintended bias. What if a property that was filtered out truly didn't have any claimable damage? Maybe it was a new property or made of superior materials compared to its neighbors. I understand that you're trying to filter out what could potentially be bad data, but you may be filtering out interesting situations.

The section's readability has been improved to better express the purpose of the filtering. A note was added that reads: One of the reasons to exclude properties without damage within an area with significant damage is because hailfall can be only a small portion of a 1 km square grid; including undamaged properties would artificially lower the actual damage created by hail when averaged. We do recognize that this method might prevent the identification of certain properties that, due to their low vulnerability, do not record damage even when exposed to hail.

Figure 4: Is there a reason why only the blue circle radars are labeled, while Table 1 includes several others that were seemingly included in the study? Seems like all that were included should be labeled in the figure.

A note in the figure caption was added that reads: The grey S-band radars recorded no hail events occurring within their domains and were not used in most of the study except section 6 and in the subsection titled "HDE - MESH relationship" as these sections do not involve claims.

A similar note was added to the section detailing the radar data (section 3).

Line 110: You mention several other data types in this section, so maybe "Radar data" isn't the best section title?

The section name was changed to "Radar Data, hail estimate products, and associated meteorology"

Line 121-123: It's unclear why this statement about the ERA5 data is included here. Please provide a connection to the SHI. I found more information about this in lines 133-134, so maybe it does not need to be duplicated in 121-123?

The following sentences were added to the SHI subsection to clarify the connection between ERA5 and SHI:

In order to calculate SHI, knowledge of the height of the 0°C and -20°C dry bulb levels are needed, as the integration of reflectivities is done only between these heights.
This information about the temperature profile was retrieved from ERA5 (Hersbach et al., 2020) from the grid point closest in time to the observation and location to the radar location.

Line 130-131: Can you please provide more explanation as to why you're doing +/- one day while in the event identification you're doing +/- seven days? I'm actually not quite following why you need the +/- one day on the SHI data, since it should have far more temporal accuracy than the damage data.

The following explanation was added to the subsecion:
The event SHI was computed for each grid point within a radar's coverage for the event's day ± one day to account for possible discrepancies between the reporting time of damage, which has at most a daily accuracy, and the time of radar observations.
This extra day on both sides of the event's date prevents situations when a hailstorm swath occurs between two days, and therefore be in the next day (or previous day, depending on which day people report most claims), and therefore only part of the swath would be used.
Since the maximum SHI for each grid point is being taken, and no two events occur one after the other in the dataset, the addition of the two extra days has no detrimental effect but solves any of the aforementioned timing issues.

The ± 7 days window on the claims site is due to the observation that events show increased claim counts above the baseline starting around one week before and returning to baseline about one week after the peak, potentially due to mislabeling of the date or errors in the reported date of damage. This explanation is in the Event Identification subsection.

Line 162-163: While not stated, you are also assuming that hail size is the controlling factor for damage. While it is a factor, it is not the only one—property age, materials, sheltering, etc. also play a role. It is OK to make assumptions, but the stipulations of those assumptions must be clearly stated.

You are correct that property age, materials, sheltering, etc. also play a role; which is why we are correcting/mitigating for such effects in the previous "Archetype normalization" subsection. When these damage grids are matched to SHI grids, the damage grid is already normalized as described in the previous section. The second paragraph of the "VA Algorithm and assumptions" subsection explicitly states that the normalised gridded damage dataset was used. We edited the starting sentence of the section to further make this point clearer as shown below:
"We assumed that the highest damage within a local area corresponded to the highest observed SHI values aloft, given that differences in vulnerabilities of properties is already mitigated by the archetype normalization."

Lines 210-211: Again noting that the assumption that the damage is only related to meteorological factors is not correct. Property factors will also play a role.

Added the following sentence to direct the reader to the subsection that addresses this point:

"since differences in vulnerabilities due to property factors have already been mitigated (see Archetype normalization subsection)."

Technical Corrections:

Line 28: reference should be in parenthesis.

Corrected, thanks

Line 35: add "and" before "therefore".

Corrected, thanks

Line 42: the information about the second limitation is fairly far removed from the original instance in the paragraph, so it may be worth restructuring this sentence to remind the reader of what the limitation is, something like "the second limitation of a mismatch between the radar-estimated hail location and actual ground observations can be mitigated by…"

The paragraph was restructured such that it reads better, thanks

Line 47: add "and" before "low-level".

Corrected, thanks

Line 69: remove closed parenthesis. In addition, you state that the ratio is just referred to as "damage", yet in Figure 1 you have "relative damage". Are these intended to be one in the same? If so, please choose one and be consistent with it.

Parenthesis removed, thanks. Also changed the y-axes label to damage.

Figure 1: label the x-axis.

Corrected, thanks

Line 92: remove "g" from "withing" and change "where" to "were".

Corrected, thanks

Line 103: "was" should be "were", and "match" should be "matched". I also think "radar's" should be "radars'" (radars followed by apostrophe) since multiple radars seem to be used?

Corrected, thanks

Line 104: should this be "the borders" instead of "a borders"? And should "radar's" be "radars'"?

Corrected, thanks

Line 105: "ares" should be "areas"?

Corrected, thanks

Line 127: add "the" before "Murrillo".

Corrected, thanks

Line 145: change "these were" to "including".

Corrected, thanks

Line 154: change "is" to "its".

Corrected, thanks

Line 159: add a period at the end of the sentence.

Corrected, thanks

Lines 171-177: you've switched to present tense in this section, while all others seem to be in past tense. Update these lines accordingly.

Corrected, thanks

Line 172: I think the end of this line needs to be adjusted to something like "allows for correction for…".

Corrected, thanks

Figure 6 caption: I'm not sure what the last sentence in the caption is for? I don't see a dashed ring?

This was needed in a previous version of the image which was zoomed out, removed. Thanks

Line 191: add "the" before "Brook".

Corrected, thanks

Line 213: seems like a word such as "so" needs to be included after the comma.

Corrected, thanks

Lines 221-222: check line formatting.

Corrected, thanks

Line 249: add "and" before "this".

Corrected, thanks

Line 252: add "the" before "VA".

Corrected, thanks

Line 275: this sentence should be restructured to something like: "due to the inability of low MESH values to cause any…". As is written now, I think values would need to be possessive (values') which is a bit awkward.

Corrected, thanks

Line 320: second "by" should be changed to "and".

Corrected, thanks

Line 325: add "and" following the comma.

Corrected, thanks

Line 328: add "the" before "ground".

Corrected, thanks

Line 331: add "a" before "higher".

Corrected, thanks

Line 366: should "mix" be "mixed"? Or "of a mix"?

Corrected to mixed, thanks

Line 369: add "and" before future.

Corrected, thanks

Line 372: change "showing" to "determine".

Corrected, thanks

Note on the training of the neural network:

We identified that our initial approach to splitting the training and validation datasets could potentially lead to model overfitting due to the high correlation between the two sets. To address this concern, we adjusted our approach by performing event-wise splitting of these sets. This modification helped reduce the possibility of high correlation in meteorological conditions between

the training and validation datasets. The resulting model maintained nearly the same level of performance as before, albeit now requiring 7 ensemble members instead of the initial 5. We have updated the model training section accordingly to reflect this adjustment.

---

## Author Comment (AC2)

Reviewer 2

Dear authors:

The authors built a Hail Damage Estimate method using Radar and ERA datasets. Some technological methods were adapted in generating data samples such as the filtering of claims and the virtual advection of SHI. These methods makes the sample more reliable. However one question confuse me: Nether the bulk advection or virtually advected were needed, but the advection algorithm needs the damage dataset as input, was this a conflict?

We apologise for the confusion, the virtual advection of MESH was absolutely needed for the analysis, as it provided the dataset used to match damage at ground with MESH/SHI aloft. It formed the basis for the training dataset (in addition to meteorological parameters) of the neural network. The bulk advection is not used directly in the neural network, but is used to qualitatively evaluate the performance of the virtual advection method.

The last paragraph in the introduction section was changed such that the way the data was used is more clear. The paragraph reads as below:

"

In our study, we leverage MESH data from the BoM's national network of weather radars, combined with a 10-year dataset of hail damage insurance claims provided by Suncorp Group Limited (Suncorp) and meteorological data from ERA5 reanalysis (Hersbach et al., 2020), to train a deep neural network capable of predicting hail damage. The structure and data flow of the study is:

      section 2: Describes the insurance data, applied filtering, and normalization.

      section 3: Describes the radar data and calculated products.

      section 4: Describes the procedure to match the insurance data to radar observations.

      section 5: Describes the development and evaluation of the neural network driven by these matched insurance and radar data and aided by meteorological data.

      section 6: The new hail damage model is applied to the full data archive, no longer limited by insurance exposure, and the relationship between the predicted hail damage, MESH, and meteorology is discussed.

"

Major Comments:

1. In Figure 1, relative damage distribution of 12 archetypes were showed, I did not see this kind of archetype were used in the following research, this was only used for normalization? The different archetype in different areas may impact the final result.

The archetype normalization was indeed used in the following research. The following sentence was added to the end of the archetype normalization subsection to clarify that this is the case.

"The rest of the study uses the normalized damage instead of the original loss ratios, this way the effect from the different vulnerabilities from the various property types can be minimized."

2. The relative damage precentage need further explained, was 0% damage in or not in the damage sample? Lots of zero damage were scattered in the following discussions such as figure 7, 8, 9,10 and 11.

The definition of damage was clarified as shown below in the Insurance Data section:
"To avoid introducing biases toward more expensive properties, we computed a damage metric known as the loss ratio, which is the ratio of incurred loss to the insured sum.
In this report, we will refer to this metric simply as damage and expressed as a percentage."

Indeed, 0% damage was included in the study. The section that describes the Claims filtering was modified to clearly state this.
"After applying all the filters, our dataset consisted of 18 intense hail events and 12 medium hail events; these provided 1,775 damage grid points, and 76,703 exposed but undamaged grid points."

3. In the claims filtering part, an obvious white cycle ring surrounding the hail area were created after filtering(Figure 2) . The discontinuous scatter round zero precentage damage (the left panel of figure 7 and bottom right in figure 10) need an explanation, was it caused by the 0% damage samples or the white cycle ring after claims filtering? The impact to CSI, POD and Far should also be discussed.

Figure 2 shows the time series of hail damage claims for Brisbane as a sample, and does not refer to filtering. We assume the reviewer is referring to figure 3. We are not certain what white cycle ring the reviewer is referring, but we speculate it might be what we refer in line 102 as a "buffer zone" between the damaged areas and non-damaged areas. Grid points in these "buffer" areas were not used in the analysis as hail damage is uncertain in this transition zones. We wanted to create a dataset with data that was as clean and certain as possible in order to minimize the uncertainty in the training dataset of the neural network.

The claims filtering section was modified to make it easier to read.

Regarding the discontinuous scatter between zero damage and about 0.5% the following paragraph was added to the manuscript in the Training and performance subsection.

"It is important to note that this CSI was achieved at 0.5\% damage threshold.
From the observed damage data (see figures 7, 8, and 9), it is evident that there are only a few claims between 0\% and approximately 0.5\%.
This is likely because losses below this ratio fall below the policies' deductibles and, therefore, are often not reported by property owners.
This apparent discontinuity in the observed damage data was also observed in the unfiltered data (Claims Filtering subsection), indicating that it is not a result of the elimination of uncertain damage areas."

4. Besides SHI, why not involve Vertical Integrated Liquid water, echo top and composite reflectivity in your machine learning model ?

Vertical integrated liquid, echo top height, and maximum column reflectivity were all included in the initial iterations of the neural network, but were eliminated from the input early on during the SHAP analysis, likely due to the high correlation to SHI. The subsection describing the neural network structure and selection of input data was modified to clearly state this.

5. What's the time interval of the new developed MSEH? In daily? Can this method be applied to instant radar datasets, which I believe more meaningful in hail warning.

It is only limited by the radar scan frequency. This is currently being developed for use as a hail nowcasting product. This possible application was added in the conclusions.

"In addition, future work will use this novel hail damage estimate for nowcasting applications to provide hail warning."

6. Some composite reflectivitymap should be showed to see if the red cycle were spurious claims in figure 11.

This was done but we believe that the MESH daily max contours shown in figure 11 demonstrate this as the lowest contour is 20mm and is far from these claims.

The following sentences were added to better guide the reader:

"It is worth noting that this case exhibited some potentially spurious claims, as highlighted within the red circles in the panel F. All these point were tracked and found in a cluster in the map (indicated by the red circle in the panel E) close to the coast and were relatively far from the main storm swaths and could be due to the misclassification of damage cause (i.e., wind or flood damage instead of hail) in the insurance dataset."

Minor Comments:

1. (a) (b) (c) (d), should be labeled in panel figures.

Corrected figures 10, 11, and 12, thanks

2. Figure 6, what's the meaning of 'The dashed range ring is 150 km in radius'?

This was needed in a previous version of the image which was zoomed out, removed. Thanks

Note on the training of the neural network:

We identified that our initial approach to splitting the training and validation datasets could potentially lead to model overfitting due to the high correlation between the two sets. To address this concern, we adjusted our approach by performing event-wise splitting of these sets. This modification helped reduce the possibility of high correlation in meteorological conditions between the training and validation datasets. The resulting model maintained nearly the same level of performance as before, albeit now requiring 7 ensemble members instead of the initial 5. We have updated the model training section accordingly to reflect this adjustment.

---

## Author Response (AR2)

Reviewer 1

General Comments:

Very nice job revising the paper to respond to the comments of both reviewers. I had a few specific editorial comments (below)

Thank you for helping us make the manuscript better.

Specific Comments:

Line 102: Suggest adding a period after "ratios" and starting a new sentence.

Done!

Line 136: Suggest removing the contraction and changing to "It is"

Done!

Line 145: Suggest capitalizing the new words in the heading to match with major section 2, 4 heading format.

Done!

Line 171-172: The language could be improved ("and therefore be in the next day" is confusing), and you have "and therefore" twice in one sentence

Done!

Line 204: I think "Algorithm" doesn't need to be capitalized?

Done!

Line 285 and 291: spell out "six"

Done!